# Structural Equation Modeling (SEM): Childhood Aggression and Irritable ADHD Associated with Parental Psychiatric Symptoms

**DOI:** 10.3390/ijerph181910068

**Published:** 2021-09-25

**Authors:** Ruu-Fen Tzang, Yue-Cune Chang, Chuan-Hsin Chang

**Affiliations:** 1Department of Psychiatry, Mackay Memorial Hospital, Taipei 104, Taiwan; 2Department of Childhood Care and Education, Mackay Junior College of Medicine, Nursing and Management, Taipei 112, Taiwan; 3Mackay Medical College, New Taipei City 252, Taiwan; 4Department of Mathematics, Tamkang University, New Taipei City 251, Taiwan; ycchang414@gmail.com; 5Agricultural Biotechnology Research Center, Academia Sinica, Taipei 115, Taiwan; chuanhsin032484@gmail.com

**Keywords:** ADHD, ODD, aggression, parental depression, structural equation modeling (SEM)

## Abstract

Background: Children with attention deficit hyperactivity disorder (ADHD) co-occurring with Oppositional Defiant Disorder (ODD) further present aggressive behavior and may have a depressive parent. A child with co-occurring ADHD and ODD has differentially higher levels of behavioral and emotional difficulties. Little is known about how the irritable subtype of ADHD in children mediates the development of parental symptomatology. This study aims to elucidate the direct or indirect influence of childhood disruptive ADHD with aggressive behavior on their parental symptom using Structural Equation Modeling (SEM). Methods: A total of 231 ADHD children and their parents completed the Swanson, Nolan, and Pelham Version IV questionnaire for symptoms of ADHD, Oppositional Defiant Disorder (ODD) scale for irritable symptoms, Child Behavior Check List (CBCL) for aggression, and Symptom Checklist (SCL) for parental symptom. Results: The three-factor confirmatory factor analysis (CFA) model found symptoms of inattention, hyperactivity/impulsivity, irritable ODD, and aggression were inter-related. Mediational analyses demonstrated ODD mediates symptoms directly predicting the risk of increasing ADHD severity. Disruptive child symptoms (ADHD + ODD + aggression) may increase the risk of depression-related symptoms in the parent. When the child’s aggression increases by one standard deviation (SD), parental psychiatric symptoms increase by 0.235 SD (*p* < 0.001). Conclusions: By this SEM pathway analysis, there is the correlation between the disruptive, more aggressive subtype of ADHD in children/adolescents and the existence of psychopathological symptomatology of their parents. ADHD + ODD + aggression in children should be classified as an irritable subtype of ADHD, warranting early diagnosis and intensive treatment.

## 1. Introduction

Attention deficit hyperactivity disorder (ADHD) is the most common neurodevelopmental disorder, with a reported high prevalence of 5% to 12.76% among children and adolescents [1]. An effective combination of pharmacotherapy with cognitive behavior therapy should be provided to these patients [2]. However, it is well known that families with hyperactive children usually are stressful [3,4]. Families with ADHD children experience more stress due to their children’s unmet needs, such as under-treated ADHD, which worsens behavior problems possibly due to social stigma problems [5,6]. In addition, parents sometimes refuse pharmacotherapy for their child [7]. All these unmet needs for support might lead parents living with ADHD children to experience different degrees of parenting stress [8]. Among the studies on parenting stress, Johnston et al. demonstrated parents living with ADHD children that have noncompliant behaviors experience more parenting stress [9]. However, regarding research on the association between ADHD and parenting stress, little is known about whether the child’s condition, such as the irritable subtype or disruptive symptom of ADHD, uniquely influences parental depression during parent–child interaction. There is a need to understand parental psychiatric symptoms when analyzing childhood ADHD symptom severity.

By the developmental-transactional model, it is reasonable to find the cause-and-effect relationship between ADHD in children and maternal depressive disorder through multiple psychosocial and biological risks [10]. As an earlier study indicated, parents with ADHD children experienced various biological and mental disorders, such as the parent suffering from more mental disorders [11], more alcohol abuse [12,13], greater likelihood the parent also has ADHD [14], and more anxiety or depression [15,16]. A common concept is that ADHD children may live with parents with psychiatric symptoms [17]. However, some studies waive this general concept, as ADHD children may have depressive mothers [18]. To date, there is no study that associates ADHD in children, particularly the irritable and aggressive subtype of ADHD, with the development of parental depression-related symptoms. 

Not every family has a difficult time. When reviewing ADHD symptom severity, we found a clinically heterogenous picture, especially when ADHD is complicated by extensive comorbid conditions [19]. Up to 80% of children with ADHD commonly report the irritable subtype [20], so-called ADHD with oppositional defiant disorder (ODD). Symptoms of ODD include disruptive, irritable, negativistic, uncooperative, defiant, and/or disobedient attitudes [21]. Childhood ODD aggravates the severity of ADHD [22,23]. In addition, symptoms of ODD are closely associated with childhood aggressive behavior [24]. According to earlier research, aggressive behavior in ODD children implies a greater risk of worse symptom severity [25,26]. Therefore, the symptom profile in ADHD, ODD, and any kind of childhood aggression could be regarded as psychopathological spectrum disorder. 

Previous studies on psychopathological spectrum disorders have focused more on the symptom profiles of ADHD, ODD, and Conduct Disorder (CD) [27], not aggression in children with irritable ADHD. The usual problem is CD may not be easily diagnosed in younger children. Younger ADHD children frequently present with aggressive behavior, such as frequent lying and destructive behavior. Children with co-occurring ADHD and ODD have differentially higher levels of behavioral and emotional difficulties. Aggression with ADHD and ODD might further increase the highest levels of behavioral and emotional difficulties in children [25,26] and also suggests recognizing early psychopathological spectrum as encompassing ADHD, ODD, and aggression. For families living with children with irritable ODD and ADHD and presenting only some aggression, more conflictual parent–child relationships were still reported [28]. Do parents with children that have irritable or hurtful ODD and ADHD, presenting different levels of aggression, have depression? Pathway analysis of ADHD symptom dimensions is needed to elucidate whether greater ADHD symptom severity, combined with ODD and aggression, may associated with parental depression-related symptoms. This pathway analysis can be used to observe the shared genetic loading between children and their parents and further help clinicians to recognize severe symptoms in ADHD children and provide more effective treatment to help parents with parental depression.

Structural equation modeling (SEM) is an appropriate method for analyzing a set of interactive factors simultaneously [29]. Traditional regression or correlation modeling may not be useful for exploring the direction of causality or mediating risks of symptoms of ADHD, ODD, aggression, and parental psychiatric symptoms simultaneously. This study utilizes SEM, uncovering the paths between ADHD (inattention and hyperactivity), disruptive child symptoms (ODD and aggression), and parental psychiatric symptoms (somatization, obsessive–compulsive disorder, interpersonal sensitivity, depression, and anxiety). SEM data elucidate whether ODD is a mediating risk factor that causes children with ADHD to become aggressive. Therefore, we can prove the association of shared genetic loading between the child and their parent.

The authors first hypothesize a bidirectional association between the symptoms of the child and parental symptoms. Following this, we examined the direct effect between ADHD, ODD, aggression, and parental symptoms. This allows us to further examine whether disruptive symptoms in children have a more direct effect on parental depression-related symptoms compared to families with kids that only have ADHD. To discover the relationship between ADHD, ODD, and aggression, the authors also hypothesized ODD is a mediating risk factor that causes children with ADHD to become aggressive and has a significant, direct effect on parental depression-related psychiatric symptoms. With these results, mental health professionals can take steps earlier to manage ODD and develop more intensive parenting programs to prevent further occurrences of conduct disorder in the future.

## 2. Materials and Methods

### 2.1. Participants and Data Collection

This study recruited patients from the outpatient unit of a major medical center in Taipei, Taiwan. The Mackay Memorial Hospital Institutional Review Board (IRB) approved the design of the study, with IRB No. MMH-I-S-489 under the project “exploring the symptomatology on children with internet addiction and attention deficit hyperactivity disorder and their parent”. After receiving a complete description of the study, participants provided written informed consent in line with the IRB guidelines. A total of 231 children with a clinical diagnosis of ADHD, based on the Diagnostic and Statistical Manual of Mental Health Disorders, Fourth Edition (DSM-IV), were recruited in this study. The senior child and adolescence psychiatrist confirmed the clinical diagnosis of ADHD based on the DSM-IV criteria. After obtaining signed consent from the legal guardian, each subject recruited for this study was invited to participate in the following programs and was interviewed to derive the following measures.

### 2.2. Measurements

#### 2.2.1. Symptoms of ADHD and ODD

The Swanson, Nolan, and Pelham version IV scale, Chinese (SNAP-IV-C) consists of the following items: inattention (9 items), hyperactivity/impulsivity (9 items), and oppositional symptoms (8 items). These items reflect the core symptoms of ADHD and ODD, as defined in DSM-IV [30]. The inattention, hyperactivity and oppositional symptom subscales of SNAP-IV-C had retest reliability (intraclass correlation coefficients (ICC)) of 0.72, 0.67, and 0.59, respectively for parent form. The ICC of three subscales of SNAP-IV-C by teacher form is from 0.60 to 0.84. All subscales by both the parent and teacher forms revealed excellent internal consistency with Cronbach’s α greater than 0.88 [31].

#### 2.2.2. Aggressive Behavior

We used the Child Behavior Check List (CBCL) to determine competencies and behavioral/emotional problems of children aged 4–18 years. The questionnaires, completed by the parents, included 118 items to assess specific behavioral and emotional problems. The CBCL was translated into Chinese via a two-stage translation [32]. The internal consistency and 1-month test–retest reliability were satisfactory for CBCL of Chinese version for child and adolescent (all α values and reliabilities > 0.6, except for thought problems) [33]. The present study analyzed the following six scales: aggressive behaviors, attention problems, anxiety/depression, social problems, delinquent behaviors, and somatic complaints. For this study, the score for aggressive behavior indicated the presence of one symptom of conduct disorder (CD). 

#### 2.2.3. Disruptive Child Symptoms (DCS)

The ODD score from SNAP plus an aggressive behavior score from CBCL was used to indicate Disruptive Child Symptoms (DCS) and to differentiate general irritable ODD or aggressive symptoms from symptoms of ADHD. 

#### 2.2.4. Parental Symptoms Checklist (SCL)

We used the Symptom Checklist-90-Revised (SCL-90-R), a 90-item self-report system developed in the 1980s by Derogatis (Derogatis, 1975), to determine the parental psychiatric symptom score. The SCL-90-R has previously been proven as a reliable psychometric measurement. The Cronbach α coefficient range of the SCL-90-R is 0.77–0.90, according to the study by Derogatis [34]. The Chinese version of this scale has been widely applied in both psychiatric ADHD studies [35] and non-psychiatric clinical studies [36] in Taiwan.

The nine primary dimensions are somatization, obsessive–compulsive disorder, interpersonal sensitivity, depression, anxiety, hostility, phobic anxiety, paranoid ideation, and psychoticism. We used a five-point scale (0–4) for symptoms, ranging from “not at all” to “extremely”. We chose five parental symptoms (somatization, obsessive–compulsive disorder, interpersonal sensitivity, depression, anxiety) for this SEM analysis to explore the association between parental depressive/anxiety symptoms and disruptive behavior symptoms of children.

### 2.3. Statistical Analyses

SEM can be used to evaluate and to explain how the relationships among unobserved variables and their influence on their parental psychiatric symptom. SEM is directly adequate latent pathway analysis to elucidate the hypothesis test. This paper reviews the role of latent variables in multiple regression, factor analysis, and structural equation models. 

SEM process centers around two steps. One is validating the measurement model: accomplished primarily through confirmatory factor analysis (CFA). Other is fitting the structural model: accomplished primarily through path analysis with latent variables [37].

The first SEM process was using factor loadings to specify the association between an unobservable construct (latent variables) and its theoretically related measures (indicator variables). Multiple linear regression methods were used to determine the relationships among the aforementioned latent variables and were indexed by standardized path coefficients. 

The second step is to use absolute model fit indices determining how well a priori model fits the sample data. The six following indices were used to evaluate model fit: (1) the chi-square test, χ^2^; (2) the comparative fit index, CFI; (3) the Bentler–Bonett [38] normed fit index, NFI; (4) the goodness-of-fit index, GFI; (5) the Tucker–Lewis index, TLI, which is also known as the Bentler–Bonett non-normed fit index (NNFI); (6) the root-mean-square error of approximation, RMSEA [39]. All these indices indicated a proposed fit of SEM data. A non-significant χ^2^ (*p* > 0.05), GFI and CFI greater than 0.95, TLI greater than 0.96, and RMSEA less than 0.06 each indicated a good fit between the data and the hypothesized model [40]. 

We used the following two models to establish the potential direct effect of a diagnosis of worse ADHD severity with related disruptive child symptoms (ODD and aggression) on the development of parental psychiatric symptoms (SCL-90). Among three latent variables, named ADHD, Disruptive Child Symptoms (DCS), and parental symptom (SCL), we used a three-factor confirmatory factor analysis (CFA) model to explore the existence of bivariate correlations. For the direct effect model of DCS on ADHD and parental symptoms (SCL-90), we used the SEM model to find the relationship. Furthermore, a mediational effect model, analyzed by SEM, was used to elucidate the relationship among ADHD, ODD, and aggression. We used Amos 21.0 and SPSS v.21.0 packages (SPSS Inc.) to perform all statistical analyses [37].

There are some limitations in using the SEM method. The most severe one is that all measurement variables (represented by rectangles) are normally distributed. In other words, the demographic factors, such as gender, marriage status, and socioeconomic status, etc., cannot be put into the structure part to control for their potential influence on the relationships in the modeling.

## 3. Results

Overall, 231 eligible patients with ADHD were enrolled. In Table 1 the numbers, means, and standard deviations of all indicator variables are listed. There were 158 ADHD patients with the combined inattentive and hyperactivity/impulsivity subtype (68.7%). The comorbidity rate of ADHD children was 73.0%. Table 2 shows the zero-order correlations of the indicator variables. 

The results of the three-factor CFA, in Figure 1, show that all indicator variables were reliable and valid measures of their respective latent variables, supported by a significant moderate-to-high factor loading (β = 0.63–0.94, *p* < 0.001). However, the covariance between perceived ADHD symptoms in children and SCL-90 parental symptoms was not significantly different from zero at the 0.05 significance level (*p*-value = 0.115). Based on the aforementioned six indices, this model provides a good fit for the data, as suggested by the results of the non-significant chi-square test and the other five indices of good fit (χ^2^ = 21.00, df = 22, *p* = 0.52, CFI = 1.0, NFI = 0.99, GFI = 0.98, TLI = 1.00, RMSEA < 0.001). 

Figure 2 shows the SEM model, positing the direct effect of DCS on ADHD and SCL-90 parental symptoms. The DCS encompassing ODD and aggression showed a significant, positive association with ADHD (β = 0.85, *p* < 0.001). In other words, when DCS increased by one standard deviation, ADHD significantly increased by 0.85 standard deviations. This SEM model also provided a very good fit for the data (χ^2^ = 21.83, df = 23, *p* = 0.53, CFI = 1.0, NFI = 0.98, GFI = 0.98, TLI = 1.00, RMSEA < 0.001).

The mediational effect model analyzed by SEM is shown in Figure 3. ODD mediated the influence of ADHD symptoms on the presentation of aggression. That is, ODD had a significant, positive association with ADHD symptoms (hyperactivity and inattention) (β = 0.71, *p* < 0.001) and is predictive of aggression in children (β = 0.31, *p* < 0.001). On the other hand, children’s aggression had a significant, direct effect on pa-rental somatization, obsessive–compulsive disorder, interpersonal sensitivity, depression, and symptoms of anxiety. More specifically, the standardized direct effect of a child’s aggression on parental symptom was 0.235 (*p* < 0.001). That is, when aggression in children increased by one standard deviation, parental psychiatric symptoms in-creased by 0.235 standard deviations. The mediation model also provided an excellent fit for the data (χ^2^ = 16.29, df = 23, *p* = 0.84, CFI = 1.0, NFI = 0.99, GFI = 0.99, TLI = 1.00, RMSEA < 0.001).

## 4. Discussion

Our study used exploratory SEM to elucidate the association between ADHD, ODD, and aggression in children with parental psychiatric symptoms. We found a potential bi-directive influence between symptoms from children with ADHD and parental depression-related symptoms. This Confirmatory Factor Analysis (CFA) conceptualized ADHD, ODD, and aggression in children interact with parental depression-related symptoms. 

We have known that adolescents with ADHD have more family-related impairments due to higher ADHD severity and parenting stress [41]. This study further indicated high ADHD severity, in combination with ODD and aggressive behavior, from the child’s perspective is interrelated with parental psychiatric symptoms (somatization, obsessive–compulsive disorder, interpersonal sensitivity, depression, anxiety). These ADHD children with more disruptive behavior symptoms predict more parental depression-related symptoms. 

According to this study result, so-called higher ADHD severity is a form of ADHD in children that co-occurs with disruptive behavior symptoms (ODD and aggression). Importantly, these three behavior symptoms (ADHD, ODD, and aggression) are inter-correlated with each other, with a ratio of 0.86 (*p* < 0.001). ODD, being a serious intermediate diagnosis, plays a role in increasing the severity of hyperactive/impulsive symptoms of ADHD. The symptoms of ADHD were not significantly associated with parental depression-related symptoms. However, if ADHD co-occurs with ODD and aggressive behavior, the inter-relevance of symptoms of the child and parental depression-related symptoms became significant, 0.23 (*p* = 0.006). Higher ADHD severity with disruptive behavior problems in children, or so-called the psychopathologic spectrum, could be used to predict the depression- and anxiety-related psychiatric symptoms in their parents, according to this SEM analysis.

This SEM pathway analysis extends our understanding of childhood aggressive behavior in ADHD with ODD and provides a new approach for assessing ADHD heterogeneity focused on the symptom profile. Here, aggressive behavior was indicated by any aggressive behavior from the CBCL symptom checklist. Although child psychiatrists are quite sure only children with ADHD + ODD + conduct disorder (CD) need an intensive program combining pharmacotherapy and parenting programs [42], here our SEM finding is congruent with one past result in a traditional regression analysis, demonstrating childhood aggressive behaviors was the only marker of increasing ADHD severity [22,23,43]. Aggressive behavior increases the loading of polygenetic risk for ADHD and symptom severity [44]. Pardini et al. suggested recognizing children with ADHD + ODD + aggression as a distinct heterogeneous group because unrecognized or under-treated childhood aggression symptoms might develop into adulthood and lead to pathologic traits in the future [45]. In addition, clinicians should screen and treat these complicated irritable ODD and aggressive symptoms early on, even using pharmacotherapy in cases of severe symptom intensity [24]. Thus, we have to pay more attention to children with more severe ADHD, already having ODD, and with any kind of aggressive behavior, and children and their parents may need to receive earlier intensive treatment together.

Another contribution of our study is the zero-order correlation, which highlighted aggressive behavior as closely correlated with the severity of hyperactivity/impulsivity symptoms (hyperactivity/impulsivity, 0.423) and with ODD (0.587) in children. As other studies have found, the severity of the children’s hyperactivity/impulsivity symptoms, but not inattention in ADHD, influence parenting stress [46]. This SEM pathway analysis also found the severity of hyperactivity/impulsivity symptoms of ADHD and ODD in children was associated with parental psychiatric symptoms. Irritable ODD is a key mediator in increasing the symptom severity of ADHD, children’s perceived comorbid aggression levels, and parental psychiatric symptoms. Therefore, ODD requires early screening and pharmacotherapy [24]. This study emphasizes the early control of symptoms of ODD and aggression in children with the irritable subtype of ADHD. Past studies have shown that children with ODD facilitate adult depression [47,48,49]. Here, we indicate treating ODD is important enough to prevent adult depression. Presently, treating irritable ADHD children with pharmacotherapy by psychostimulant (Methylphenidate) or Atomoxetine alone is not enough. Under the influence of developmental transactions, families with children who have severe emotional disorders need combined pharmacotherapy with parental cognitive behavior therapy. Therefore, the clinician should provide knowledge of accurate pharmacotherapy and other psychological and social interventions for parents with children that have the irritable ODD subtype of ADHD. For children with ADHD/ODD and severe aggressive behavior, pharmacotherapy should focus not only on stimulant use but also other antipsychotics drug to control their aggression [50]. Furthermore, ADHD parenting training programs for families with unusually aggressive children should focus on the management of their children’s aggressive behavior. 

This study has the following limitations. First, a symptom-by-symptom checklist (SCL-90) was used to assess parental psychiatric status, rather than directly interviewing parents. Therefore, construction of the subscale seems to be arbitrary in symptom expression. Another limitation is the cross-sectional design, which may not necessarily represent the longitudinal developmental relationship among ADHD, ODD, aggression, and parental depression-related symptoms. In addition, most of the information was from the primary caregiver (mainly mothers), which may lead to sampling deviations. Finally, since the objective of this study was to explore the relationship between symptom severity in children and parental depression-related symptoms, we used the presence of any aggressive behavior in the CBCL, instead of a formal diagnosis of Conduct Disorder, which is a well-known diagnosis of childhood disruptive behavior symptoms. Evaluating aggressive behavior symptoms in the outpatient setting may be particularly challenging for some families because parents may refuse to report their child’s aggression symptoms during interview. Despite these limitations, we still suggest using the CBCL aggression scale as a tool to screen for these high-risk patients [51].

## 5. Conclusions

This study used SEM to conceptualize the intergenerational transmission of ADHD in children and parental psychiatric symptoms. We first provide this view to understand how ODD moderately enhances the severity of ADHD symptoms. In summary, this study conceptualized symptoms of ADHD, ODD, and aggression interact with parental depression-related symptoms. It is important and requires low costs to routinely screen ADHD children with ODD and aggressive behavior using the CBCL in a clinical setting. ADHD children on the severe end of the psychopathologic spectrum require a more intensive biopsychosocial model of treatment to help them earlier in clinical psychiatric practice [52].

## Figures and Tables

**Figure 1 ijerph-18-10068-f001:**
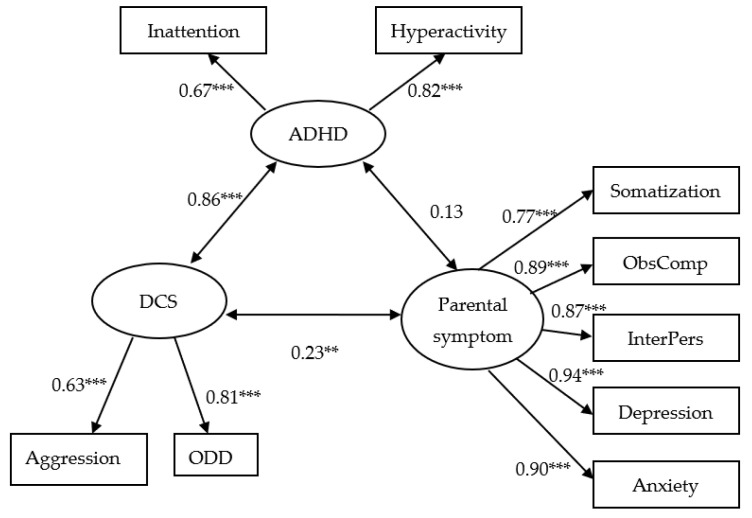
Three factor’s confirmatory factor analysis (CFA) model (positing the existence of bivariate correlations among three latent variables: DCS, ADHD, and Parental symptom). Circles represent unobserved latent variables. Rectangles represent observed measured variables. Values are standardized path coefficients. DCS: Disruptive Child Symptom; ObsComp: Obsessive compulsive; InterPers: Interpersonal sensitivity. ***: *p* < 0.001. **: *p* < 0.01. This SEM model also provided a very good fit for the data (χ^2^ = 21.00, df = 22, *p* = 0.52, CFI = 1.0, NFI = 0.98, GFI = 0.98, TLI = 1.00, RMSEA < 0.001). CFI: Comparative Fit Index; NFI: Normed Fit Index; GFI: Goodness-of-fit; RMSEA = Root Mean Squared Error of Approximation.

**Figure 2 ijerph-18-10068-f002:**
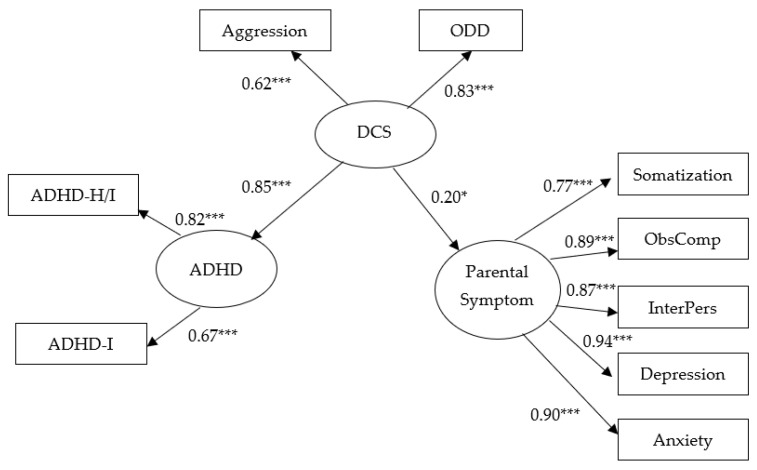
The SEM model (positing the direct effect model of DCS on ADHD and SCL). Circles represent unobserved latent variables. Rectangles represent observed measured variables. Values are standardized path coefficients. DCS Disruptive Child Symptom; ObsComp: Obsessive compulsive; InterPers: Interpersonal sensitivity. ***: *p* < 0.001. *: *p* < 0.05. This SEM model also provided a very good fit for the data (χ^2^ = 21.83, df = 23, *p* = 0.53, CFI = 1.0, NFI = 0.98, GFI = 0.98, TLI = 1.00, RMSEA < 0.001). CFI: Comparative Fit Index; NFI: Normed Fit Index; GFI: Goodness-of-fit; RMSEA = Root Mean Squared Error of Approximation.

**Figure 3 ijerph-18-10068-f003:**
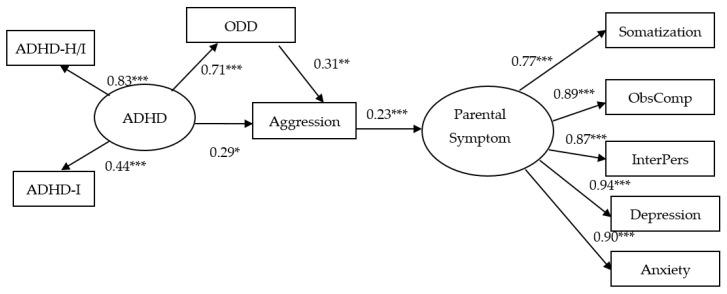
The SEM mediational effect’s model (ODD was a mediator to the direct effect of ADHD symptom to aggression which had a significant direct effect on parental symptom). Circles represent unobserved latent variables. Rectangles represent obScheme 0. ***: *p* < 0.001. **: *p* < 0.01. *: *p* < 0.05. This SEM model also provided a very good fit for the data (χ^2^ = 16.29, df = 23, *p* = 0.84, CFI:1.0, NFI = 0.98, GFI = 0.96, TLI = 1.008, RMSEA < 0.001). CFI = Comparative Fit Index; NFI: Normed Fit Index; GFI: Goodness-of-fit; RMSEA = Root Mean Squared Error of Approximation.

**Table 1 ijerph-18-10068-t001:** Sample characteristics of study measures.

Characteristics		N	Mean, %	SD
Age		231	10.17	2.59
Male (%)		175	75.8%	
Comorbidity	Yes	168	73.0%	
	No	62	27.0%	
Subtype	Combined	158	68.7%	
	Inattentive	72	31.3%	
Education	Elementary school	171	75.0%	
	Junior high school	54	23.7%	
	Senior high school	3	1.3%	
ADHD				
	Inattention	231	17.19	4.50
	Hyperactivity	231	12.43	6.46
Disruptive child symptom				
	ODD	231	12.25	5.82
	Aggression	231	13.32	7.23
Parental symptom (SCL-90)				
	Somatization	231	4.53	6.19
	Obsessive compulsive	231	5.68	5.53
	Interpersonal sensitivity	231	3.31	4.10
	Depression	231	5.11	6.08
	Anxiety	231	2.54	3.43

SD, Standard Deviation; ODD, Oppositional Defiant Disorder.

**Table 2 ijerph-18-10068-t002:** Zero-order correlations among study measures.

	1	2	3	4	5	6	7	8
1. Inattention	-							
2. Hyperactivity	0.553 **	-						
3. Aggression	0.198 **	0.423 **	-					
4. ODD	0.469 **	0.587 **	0.515 **	-				
5. Soma (P)	0.070	0.015	0.127	0.052	-			
6. ObsComp (P)	0.137 *	0.074	0.235 **	0.132 *	0.663 **	-		
7. InterPers (P)	0.150 *	0.076	0.218 **	0.135 *	0.528 **	0.784 **	-	
8. Depression (P)	0.128	0.048	0.210 **	0.124	0.727 **	0.825 **	0.822 **	-
9. Anxiety (P)	0.141 *	0.082	0.232 **	0.121	0.697 **	0.806 **	0.758 **	0.849 **

Soma, Somatization; P, parent; ObsComp, Obsessive compulsive; InterPers, Interpersonal sensitivity; **: *p* < 0.001; *: *p* < 0.05.

## Data Availability

Not applicable.

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
