# Peer review of "Structural Equation Modeling (SEM): Childhood Aggression and Irritable ADHD Associated with Parental Psychiatric Symptoms"

_ijerph, 2021, doi:10.3390/ijerph181910068_

Round 1

Reviewer 1 Report

The manuscript provides an in-depth analysis of the correlation between symptoms of ADHD and/or ODD in a child and parental risk of developing a psychiatric disorder.

The paper is interesting. I have however a few points to raise.

1) The English of the paper is not always good, some sentences are hard to follow.

2) Was a priori sample size calculated? How or why not?

3) Why were only some scales of CBCL and SCL-09-R used for the analysis?

4) Why were DSM IV criteria and not DSM 5 criteria used?

5) Table 1 is a bit hard to understand. It does not provide informations about how many subjects have different subtypes of ADHD.

Author Response

Response to Reviewer 1 Comments

The manuscript provides an in-depth analysis of the correlation between symptoms of ADHD and/or ODD in a child and parental risk of developing a psychiatric disorder.

The paper is interesting. I have however a few points to raise.

Point 1: The English of the paper is not always good, some sentences are hard to follow.

Response 1: Thank you for suggestion. We will further check some sentence one by one. Indeed, we have invited MDPI's English editing service to work through our English expression.

Point 2: Was a priori sample size calculated? How or why not?

Response 2: The aim of this study is using SEM pathway analysis to explore the correlation between symptoms of ADHD and/or ODD in a child and parental risk of developing a psychiatric disorder. We did not design randomized controlled trials (RCTs) or priori sample size calculation. Also, according to the experiments and studies of SEM, the minimum sample size is 200 cases. Our study sample size is 231 more than minimum sample size.

Point 3: Why were only some scales of CBCL and SCL-09-R used for the analysis?

Response 3: Thank you for suggestion. SEM is a pathway analysis to elucidate how the ODD mediated the relation between ADHD to aggression and parental risk of developing a psychiatric disorder. In SEM, here ADHD is latent variable; ODD and aggression is measurement variable. Beside we used CBCL to check childhood aggressive behavior problem and SCL-90-R to check the parental psychiatric symptom. We also used SNAP to check symptoms of ADHD and/or ODD. Then we used pathway analysis-SEM to get the correlation between symptoms of ADHD and/or ODD in a child and parental risk of developing a psychiatric disorder.

Point 4: Why were DSM IV criteria and not DSM 5 criteria used?

Response 4: DSM-5 criteria were first published on 2013 and updated on 2015 in The States. The psychiatrist in Taiwan had to wait for the DSM-5 criteria to be translated then started to use it later. Therefore, in Taiwan we were still using the DSM IV criteria, when we were collecting cases for this research.

Point 5: Table 1 is a bit hard to understand. It does not provide information about how many subjects have different subtypes of ADHD.

Response 5: Thank you for your suggestion. We have checked the Table 1 to become readable. Inside Table 1, we had information of combined and inattentive subtype. That is 68.7% (158/231) of them is combined subtype and 31.1% (72/231) is inattentive subtype of ADHD.

Reviewer 2 Report

This is an interesting and novel paper dealing with the relation between the disruptive, more aggressive  subtype of ADHD in children/adolescents and the existence of psychopathological symptomatology (emphasis given on depressive symptoms) of their parents.

Despite the interesting topic, there are major issues in the methodology of research (at least the way it is presented) that reduce its final validity:

  1. The  abstract needs restructuring, since the expected sections of background, methods, results, conclusion  are not discernible in the text rendering the understanding of the reader difficult.
  2. It is not obvious from the main text how parental cognitive behavioural  interventions mentioned in the last sentence of the abstract may affect reduction of parental depressive symptomatology. This is not part of the research hypothesis of the paper and not substantiated by further references even in the discussion section.
  3. Methodology is not clearly defined:
  4. Where the children and adolescent participants recruited in a row as they presented to the external department of the ADHD clinic? Were they randomly selected? Were there any drop-outs in the process?
  5. Who completed the questionnaires on ADHD? was it the children adolescents themselves after having the consent of their legal custodian as stated in lines 116-117, or was the information given by the parent/teacher as stated in the limitations of the study in lines 294-295?
  6. Concerning parental participation: No information is given about age, sex, marital status ( socio-economic status neither) of participating parents. One has to go through the text again to understand that mainly mothers (lines 294-295) participated in this research, but also teachers (?) gave their opinion on measures concerning the children/adolescents? 
  7. SCL-90 is a reliable element on detecting adult psychopathology. However, it is not clear from the text, who were these parents who filled it out (mothers/fathers; single parents; younger, older; wealthy/poor) rendering any correlation to the disruptive side of their children's ADHD unreliable due to the many confounding variables intervening in a cross-sectional ( as stated) depiction of depressive symptomatology.

Author Response

Response to Reviewer 2 Comments

Comments and Suggestions for Authors

This is an interesting and novel paper dealing with the relation between the disruptive, more aggressive subtype of ADHD in children/adolescents and the existence of psychopathological symptomatology (emphasis given on depressive symptoms) of their parents.

Despite the interesting topic, there are major issues in the methodology of research (at least the way it is presented) that reduce its final validity:

Point 1: The abstract needs restructuring, since the expected sections of background, methods, results, conclusion are not discernible in the text rendering the understanding of the reader difficult.

Response 1: Thank you for your suggestion. we will further structure the abstract according to expected sections of background, methods, results, conclusion.

Point 2: It is not obvious from the main text how parental cognitive behavioral interventions mentioned in the last sentence of the abstract may affect reduction of parental depressive symptomatology. This is not part of the research hypothesis of the paper and not substantiated by further references even in the discussion section.

Response 2: Thank you for your suggestion. We will delete this part as reviewer’s suggestion. Also, we have added reviewer’s comment as There is the correlation between the disruptive, more aggressive subtype of ADHD in children/adolescents and the existence of psychopathological symptomatology of their parents” on the end of conclusion part of abstract” on page 1, line 24 to 26.

Point 3: Methodology is not clearly defined:

Response 3: Thank you for your suggestion. We have further checked the method part and re-arranged the text like following on page 4, line161 to 184:

SEM process centers around two steps. One is validating the measurement model: accomplished primarily through confirmatory factor analysis (CFA). Other is fitting the structural model: accomplished primarily through path analysis with latent variables.

The first SEM process is using factor loadings to specify the association between an unobservable construct (latent variables represented as eclipse) and its theoretically related measures (indicator variables represented as rectangle). Multiple linear regression methods were used to determine the relationships among the aforementioned latent variables and were indexed by standardized path coefficients.

Second step is to use absolute model fit indices determining how well a priori model fits the sample data. The six following indices were used to evaluate model fit: (1) the chi-square test, χ2; (2) the comparative fit index, CFI; (3) the Bentler–Bonett [38] normed fit index, NFI; (4) the goodness-of-fit index, GFI; (5) the Tucker–Lewis index, TLI, which is also known as the Bentler–Bonett non-normed fit index (NNFI); (6) the root-mean-square error of approximation, RMSEA [39]. All these indices indicated a proposed fit of SEM data. A non-significant χ2 ( p > 0.05), GFI and CFI greater than 0.95, TLI greater than 0.96, and RMSEA less than 0.06 each indicated a good fit between the data and the hypothesized model [40].

Point 4: Where the children and adolescent participants recruited in a row as they presented to the external department of the ADHD clinic? Were they randomly selected? Were there any drop-outs in the process?

Response 4: This study recruited patients from the outpatient unit of a major medical center in Taipei, Taiwan. A total of 231 children with a clinical diagnosis of ADHD, based on the Diagnostic and Statistical Manual of Mental Health Disorders, Fourth Edition (DSM-IV), were recruited in this study. The purpose of this study is exploring the relation between the disruptive, more aggressive subtype of ADHD in children/adolescents and the existence of psychopathological symptomatology of their parents. We did not design randomized controlled trials (RCTs). Also, we had no follow up the treatment efficacy or drop-outs in the process.

Point 5: Who completed the questionnaires on ADHD? was it the children adolescents themselves after having the consent of their legal custodian as stated in lines 116-117, or was the the information given by the parent/teacher as stated in the limitations of the study in lines 294-295?

Response 5: As we know from the text of this article: “The senior child and adolescence psychiatrist confirmed the clinical diagnosis of ADHD based on the DSM-IV criteria. After obtaining signed consent from the legal guardian, each subject recruited for this study was invited to participate in the following programs and was interviewed to derive the following measures. “

Child and adolescent and their parent completed the questionnaires after we had obtained signed consent from the legal guardian. We have deleted the teacher on limitation part. Cause in our study, the primary caregiver is mainly mother not teacher.

Point 6: Concerning parental participation: No information is given about age, sex, marital status ( socio-economic status neither) of participating parents. One has to go through the text again to understand that mainly mothers (lines 294-295) participated in this research, but also teachers (?) gave their opinion on measures concerning the children/adolescents? 

Response 6: Thank you for your suggestion. We had no teacher gave their opinion on measures concerning the children/adolescents. We have deleted the teacher on limitation. Cause in our study, the primary caregiver is mainly mother not teacher.

Indeed, we have basic information of their parent not shown in Table 1. Only because the purpose of this study is to find the correlation between the disruptive, more aggressive subtype of ADHD in children/adolescents and the existence of psychopathological symptomatology of their parents. Therefore, we only show parental symptom information of SCL-90 here to perform this SEM pathway analysis.

Point 7: SCL-90 is a reliable element on detecting adult psychopathology. However, it is not clear from the text, who were these parents who filled it out (mothers/fathers; single parents; younger, older; wealthy/poor) rendering any correlation to the disruptive side of their children's ADHD unreliable due to the many confounding variables intervening in a cross-sectional ( as stated) depiction of depressive symptomatology.

Response 7: Thank you for your suggestion. In our study, the primary caregiver is mainly mother. According to this SEM study result, we indicated children’s aggression had a significant, direct effect on parental somatization, obsessive–compulsive disorder, interpersonal sensitivity, depression, and symptoms of anxiety. More specifically, the standardized direct effect of a child’s aggression on parental symptom was 0.235 (p < 0.001). That is, when aggression in children increased by one standard deviation, parental psychiatric symptoms increased by 0.235 standard deviations. The mediation model also provided an excellent fit for the data (χ2 = 16.29, df = 23, p = 0.84, CFI=1.0, NFI = 0.99, GFI=0.99, TLI=1.00, RMSEA < 0.001).

By this study result, here we highlight that child with ADHD/ODD and severe aggressive behavior, their parent may have a risk to develop psychiatric symptom. Therefore, clinician should provide more effective ADHD parenting training programs in addition to psychopharmacotherapy for families with unusually aggressive children.

Round 2

Reviewer 1 Report

The authors have answered effectively to my concerns.

Author Response

Response to Reviewer 1Comments(2nd time)

Point 1: The authors have answered effectively to my concerns.

Response 1: Thank you very much.

Reviewer 2 Report

The paper is scientifically substantiated in the part it investigates children/adolescents with  the disruptive type of ADHD/ODD.

Improvements were introduced in the text ( abstract, elaboration on methodology).

Still, little is known of the parents that participated in the study and the question remains if the parental psychopathological symptom- as depicted in SCL-90- can be validly used as a variable without being controlled for other parameters (demographics, socioeconomic status etc) that may influence depressive symptomatology anyway. Otherwise, bias may not be excluded.

Author Response

Response to Reviewer 2 Comments(2nd time)

The paper is scientifically substantiated in the part it investigates children/adolescents with  the disruptive type of ADHD/ODD.

Improvements were introduced in the text (abstract, elaboration on methodology).

Point 1: Still, little is known of the parents that participated in the study and the question remains if the parental psychopathological symptom- as depicted in SCL-90- can be validly used as a variable without being controlled for other parameters (demographics, socioeconomic status etc) that may influence depressive symptomatology anyway. Otherwise, bias may not be excluded.

Response 1: Thank you for your suggestion. For the reason why the parental symptom of SCL-90 was not controlled by other parameters. Because there are some limitations in using the SEM method. The most severe one is that all measurement variables (represented by rectangles) are normally distributed. In other words, the demographics factors, such as gender, marriage status and socioeconomic status etc. cannot be put into the structure part to control for their potential influence on the relationships in the modeling. We added this limitation in the discussion part on page 8, line 313-318.

Round 3

Reviewer 2 Report

I acknowledge your serious effort. Still, I am not sure how the presentation of the parameters concerning parental identity and their subsequent omission- as presented in the limitations- would hinder the effectiveness of  SEM model.

Moreover, in the new version emphasis is given on the diagnostic value of CBCL aggression scale in the end of the discussion and  in the conclusion section, since the relation between the aggressive ADHD/ODD subtype and parental  symptomatology is being moved to the background.

I am afraid that this development disarranges the connection of the aims of the abstract with the conclusion, with CBCL scale being mentioned in the methods section along with SCL-90 in the abstract and not as the object of research as it appears in the discussion/conclusion section.

Author Response

Response to Reviewer 2 Comments (3rd time)

Point 1: I acknowledge your serious effort. Still, I am not sure how the presentation of the parameters concerning parental identity and their subsequent omission- as presented in the limitations- would hinder the effectiveness of SEM model.

Response 1: Thank you for suggestion. For more clear explanation of SEM, we have moved this part to statistical analysis of method section on page 4, line 195-198.

Point 2: Moreover, in the new version emphasis is given on the diagnostic value of CBCL aggression scale in the end of the discussion and in the conclusion section, since the relation between the aggressive ADHD/ODD subtype and parental symptomatology is being moved to the background.

Response 2: Thank you for suggestion again, we have added “with aggressive behavior” on background part to highlight the diagnostic value of CBCL aggression on page 1, line 11.

Point 3: I am afraid that this development disarranges the connection of the aims of the abstract with the conclusion, with CBCL scale being mentioned in the methods section along with SCL-90 in the abstract and not as the object of research as it appears in the discussion/conclusion section.

Response 3: Thank you for suggestion again. We have further checked this part. The aim of this study is to elucidate the direct or indirect influence of childhood disruptive ADHD with aggressive behavior on their parental symptom using Structural Equation Modeling (SEM). Indeed, we have mentioned “the influence of childhood disruptive ADHD with aggressive behavior on their parental symptom using Structural Equation Modeling (SEM)” in background of abstract. In method of abstract, we showed CBCL scale being mentioned in the methods section along with SCL-90. In conclusion section of abstract, we have rearranged two sentences of conclusion of abstract as following on page 1, line 21-25.

Conclusion By this SEM pathway analysis, there is the correlation between the disruptive, more aggressive subtype of ADHD in children/adolescents and the existence of psychopathological symptomatology of their parents. ADHD+ ODD+ aggression in children should be classified as an irritable subtype of ADHD, warranting early diagnosis and intensive treatment.
